# Effect of Hypoxia on Glucose Transporter 1 and 3 Gene Expression in Placental Mesenchymal Stem Cells Derived from Growth-Restricted Fetuses

**DOI:** 10.3390/genes13050752

**Published:** 2022-04-25

**Authors:** Yao-Lung Chang, Shuenn-Dyh Chang, An-Shine Chao, Martin Sieber, Chia-Lung Tsai, Po-Jen Cheng

**Affiliations:** 1Department of Obstetrics and Gynecology, Chang Gung Memorial Hospital, College of Medicine, Chang Gung University, Taoyuan 333, Taiwan; gene@cgmh.org.tw (S.-D.C.); aschao1295@cgmh.org.tw (A.-S.C.); pjcheng@cgmh.org (P.-J.C.); 2BIONET Corp., No.28, Ln.36, Xinhu 1st Rd., Neihu Dist., Taipei 114, Taiwan; martinsieber@bionetcorp.com; 3Genomic Medicine Research Core Laboratory, Chang Gung Memorial Hospital, Taoyuan 333, Taiwan; cltsai24@gmail.com

**Keywords:** glucose transporter, intrauterine growth restriction, hypoxia, placenta, mesenchymal stem cells

## Abstract

(1) Background: Glucose is transferred from maternal blood to the fetus by glucose transporters. What is the effect of hypoxia on the gene expression of placenta glucose transporter 1 (*GLUT1*) and glucose transporter 3 (*GLUT3*) in growth-restricted fetus is interesting. (2) Methods: The gene expression of *GLUT1* and *GLUT3* and the protein expression of HIF-1α were evaluated under nonhypoxic conditions and after 4 and 8 h under hypoxic conditions in placental mesenchymal stem cells derived from monochorionic twin pregnancies with selective intrauterine growth restriction. (3) Results: The gene expressions of *GLUT1* and *GLUT3* under hypoxia conditions were higher in placental mesenchymal stem cells derived from appropriate-for-gestational-age fetuses than in those from selective intrauterine growth-restricted fetuses. However, the protein expression of hypoxia induced factor-1 α (HIF-1α) at hypoxia condition was not lower in placenta mesenchymal stem cells from selective intrauterine growth-restricted fetuses than in placental mesenchymal stem cells from appropriate-for-gestational-age fetuses. (4) Conclusions: Hypoxia-induced upregulation of *GLUT**1* and *GLUT3* expression was decreased in placental mesenchymal stem cells from selective intrauterine growth-restricted fetuses but not due to decreased HIF-1α expression. Selective growth-restricted fetuses have less capacity for hypoxia-induced upregulation of placental glucose transport.

## 1. Introduction

Glucose, which is essential for oxidative metabolism in the growing placenta and fetus, is transferred from maternal blood to the fetus through glucose transporters (GLUTs) [1,2]. As gestational age progresses, *GLUT1* and *GLUT3* have increasingly critical roles to play. Studies have reported that when maintained under hypoxic conditions in vitro, trophoblasts isolated from at-term placentas exhibited an increase in *GLUT1* and *GLUT3* mRNA [3,4]. We previously reported that the gene expression of *GLUT1* and *GLUT3* was detected in monochorionic (MC) twin placentas after delivery [5]; and hypoxia had be reported as it could increase the *GLUT1* and *GLUT3* expression in placenta mesenchymal stem cells (PMSCs) [6].

Intrauterine growth restriction (IUGR) is a primary cause of perinatal mortality and morbidity, affecting approximately 7–15% of pregnancies [7,8]. Under hypoxic conditions in the uterus, a fetus with restricted intrauterine growth must adapt promptly because of the compromise of placental perfusion to the fetus [9]. Hypoxic conditions have been reported as possibly leading to impaired mitochondrial function in trophoblasts in growth-restricted fetuses [10]. *GLUT1* expression was upregulated in mouse placentas near term but not in placentas with decreased uterine perfusion pressure (usually observed in growth-restricted fetuses) [11]. To date, no study has investigated changes in placental glucose transporters expression under hypoxia in growth-restricted fetuses, especially in human pregnancies.

Fetal hypoxia was reported to be associated with decreased placental perfusion but not with decreased glucose uptake [12]. Glucose metabolism and transport may continually adapt to adverse conditions in the placenta [13]. Studies of placental glucose transporters’ expression in human pregnancy affected by IUGR have had conflicting results; placenta *GLUT1* and *GLUT3* gene expression has been reported to be not different between IUGR and non-IUGR fetuses or trophoblasts isolated from term placenta that demonstrated increased *GLUT1* and *GLUT3* mRNA [14]. In clinical practice, IUGR is considered when the estimated weight of a fetus is below the 10th percentile for its gestational age in a singleton pregnancy [15]. However, estimated weight below the 10th percentile would not necessarily indicate IUGR and can be small for gestational age. In order to selectively study objects as true IUGR a twin pregnancy with selective IUGR (sIUGR) model was adopted. In a twin pregnancy with sIUGR, the growth of one of the twins is restricted, with the other twin being appropriate-for-gestational-age (AGA). Because the AGA twin can grow to normal size, the sIUGR fetus likely reflects the true growth restriction. The PMSCs GLUT expression of the AGA twin can be compared with that of the sIUGR twin. Monochorionic (MC) twins are monozygotic twins who share almost identical genes. MC twin pregnancies with sIUGR provide an ideal case for evaluating differences in placental GLUTs expression between growth-restricted and normal-growth fetuses [5].

When cells are exposed to hypoxia, a primary cellular response is the induction of the hypoxia-inducible factor (HIF). HIF is a global transcriptional regulator that controls the expression of more than 1000 proteins by binding hypoxic response elements in the gene regulatory region [16]. HIF is a heterodimer protein comprised of two subunits: an α subunit with two isoforms, HIF-1α and HIF-2α, and a β subunit. Hypoxia can upregulate hypoxia-inducible factor-1α (HIF-1α) expression in trophoblast cells [17], and up-regulation of GLUT1 and GLUT3 under hypoxic conditions was demonstrated to be mediated by the HIF-1α. Therefore, HIF-1α protein expression is measured in PMSCs. We hypothesized that hypoxia would trigger the upregulation of GLUT expression but the level of upregulation would be lower in an IUGR placenta than in an AGA placenta, making the IUGR fetus more vulnerable to hypoxia. Performing in vivo studies to evaluate human placentas under hypoxic conditions is difficult; therefore, we evaluated the response of PMSCs to hypoxia as a proxy for measuring hypoxia in an in vivo fetus.

This study investigated *GLUT1* and *GLUT3* gene expression and HIF-1α protein expression in PMSCs derived from MC twins with sIUGR and cultured under normal oxygen and hypoxic conditions; changes in placental glucose transport under hypoxia in growth-restricted and AGA-growth fetuses were evaluated to determine whether IUGR placentas exhibit lower hypoxia-induced upregulation of GLUT gene expression.

## 2. Materials and Methods

The study protocol was approved by the Institutional Review Board of the Chang Gung Medical Foundation (IRB #201802017A3). An MC twin pregnancy with sIUGR had one IUGR fetus and one AGA fetus, with a birth weight discordance of >25%. IUGR was defined as an estimated fetal weight below the 10th percentile on a standard birth weight chart for a singleton pregnancy [18]. The chorionicity of the twin pregnancy was determined in the first trimester or early second trimester by an experienced sonographic specialist and confirmed through a placenta check during delivery. Exclusion criteria were the presence of twin-to-twin transfusion syndrome, twin anemia–polycythemia sequence, monoamniotic twins, and structural or genetic malformations in the twins at delivery. Only MC twins delivered through cesarean section were included to ensure an intact placenta. Birth weight discordance was defined as [(birth weight of the AGA twin—birth weight of the sIUGR twin) ÷ (birth weight of the AGA twin)] × 100%.

### 2.1. Inspection and Collection of Placental Tissues

This study enrolled six pairs of MC twins with sIUGR; their placental tissues were obtained during cesarean section, and PMSCs were successfully cultured. Placentas were cut along the vascular equator. Regions with obvious calcification or infarction were excluded. The separated placental tissues were briefly rinsed with saline to remove blood. PMSCs were cultured from each placental territory.

### 2.2. Isolation and Culture of PMSCs

The placental tissues were washed with Dulbecco’s phosphate-buffered saline (PBS; Hyclone) to remove as much blood as possible. The tissue was diced into small pieces and digested with 300 U/mL collagenases and 100 U/mL hyaluronidases (Stem Cell Technologies, Vancouver, BC, Canada) in an α-minimum essential medium (Invitrogen) at 37 °C for 2 h. After digestion, the sample was filtered through a 100 µm cell strainer (Falcon). Cells in the filtrate were collected through centrifugation at 400× *g* for 5 min. The cell pellet was resuspended and cultured in α-minimum essential medium supplemented with 10% fetal bovine serum (FBS; Gibco) in a 37 °C incubator with 5% CO_2_. After 3 days, 3 mL of culture medium was added; after 7 days, nonadherent cells were removed, and fresh culture medium was added. The culture medium was then replaced every 3 to 4 days. Subculture was performed at a plating density of 3–6 × 10^3^ cells/cm^2^.

The culture for PMSCs entailed the following: Isolated PMSCs were seeded in six-well plates at a density of 1 × 10^5^ cells in each well. After attachment to the well, the cells were cultured under normoxic (20% O_2_ and 5% CO_2_) indefinitely or hypoxic conditions (1% O_2_ and 5% CO_2_) for 4 or 8 h. The hypoxic condition (< 1%) was achieved by flushing the chamber with 5% CO_2_ and 95% N_2._ The hypoxia exposure was confirmed with O_2_ concentration monitoring and elevation of HIF-1α protein abundance with Western blotting [19].

### 2.3. PMSC Characterization

The PMSCs were characterized through flow cytometry by using fluorescein isothiocyanate-conjugated or phythoerythrin-conjugated antibodies (BD Biosciences). For analysis, the cells were detached with 0.25% trypsin/ethylenediaminetetraacetic acid in PBS for 5 min, subsequently neutralized with culture medium containing 10% FBS, and then washed with PBS. The cells were incubated for 30 min with their respective antibodies. The presence of the following markers on the cells was examined using an FACS Calibur flow cytometer (BD Biosciences): CD13, CD14, CD29, CD31, CD34, CD44, CD45, CD73, CD90, CD105, and HLA-DR. Data acquisition with at least 10,000 events per sample was performed using Cell Quest Pro (Becton, Dickinson and Co., San Jose, CA, USA). The positive markers evaluated were CD44, CD29, CD90, CD105, CD73, and CD13. The negative markers evaluated were CD34, D45, CD14, HLA-DR, and CD31. The purity was >95% positive and <2% positive for control markers (Appendix A).

### 2.4. RNA Extraction and Real-Time qPCR

Total RNA from the PMSCs was isolated using TRIzol reagent (Invitrogen). First-strand cDNA for real-time qPCR was synthesized using the cDNA reverse transcription kit (Applied Biosystems, Life Technologies, Carlsbad, CA, USA). A similar GLUT1 and GLUT3 gene expression analysis process was described previously [5]. GLUT1 expression levels were detected using the forward primer 5′-tatctgagcatcgtggccat-3′ and the reverse primer 5′-aagacgtagggaccacacag-3′. GLUT3 expression levels were detected using the forward primer 5′-tcctgggtcgcttggttatt-3′ and the reverse primer 5′-agggctgcactttgtaggat-3′. The qPCR conditions were as follows: initial denaturation at 95 °C for 10 min, followed by 40 cycles of 95 °C for 15 s and 60 °C for 60 s, plus a final melt curve step using Power SYBR Green Master Mix (Applied Biosystems, Life Technologies, Carlsbad, CA, USA). The relative expression level of each sample was normalized to that of GAPDH. Data were calculated using a comparative Ct method by using the 2^−ΔΔCt^ formula.

### 2.5. Western Blot Analysis

The PMSCs were lysed in RIPA buffer (Millipore, MA, USA) by using proteinase inhibitors (Millipore, MA, USA). After centrifugation at 12,000× *g* rpm for 15 min at 4 °C, the protein concentration was determined using the Bradford protein assay (Biorad Laboratories, CA, USA). The lysates were subjected to sodium dodecyl sulfate–polyacrylamide gel electrophoresis (SDS–PAGE), and the separated proteins were subsequently transferred onto polyvinylidene fluoride membranes (Millipore, MA, USA). The antibodies for HIF-1α used were GTX628480, 1:1000 (Genetex, Hsinchu, Taiwan). Horseradish peroxidase-conjugated antibodies and chemiluminescence reagents were obtained from Millipore. The signal intensity of autoradiograms was quantified using VisionWorks (UVP, Analytik Jena GmbH, Germany) after normalization to the corresponding actin intensity.

Hypoxic gene (protein) expression was measured as gene (protein) fold changes (FCs); FCs were examined at 4 h (FC4) and 8 h (FC8) in the culture. An FC value of more than 1 indicated that the gene or protein expression was upregulated after exposure to hypoxia.

Statistical analysis was performed using SPSS 11.0 (SPSS Inc., Chicago, IL, USA). Data are expressed as the mean ± standard deviation and median (interquartile range) and frequency (percentage) when appropriate. Qualitative data were compared using the χ^2^ test or Fisher’s exact test (when cells had an expected count of less than five). Continuous variables were tested for normality. A two-sample Student’s t test or Mann–Whitney U test was used to compare continuous variables between groups. The Wilcoxon signed-rank test was used to evaluate PMSC gene and protein expression between AGA and sIUGR fetuses. A probability value of <0.05 was considered statistically significant.

## 3. Results

The characteristics of the six pairs of twins are listed in Table 1. The median birth weight discordance of the six pairs of MC twins was 49.7%; the median gestational age at delivery was 30.7 weeks. Two cases had reversed end-diastolic flow of the umbilical artery to the sIUGR fetus just before delivery, and one case had no flow. All six pairs of MC twin pregnancies were delivered through cesarean section on the basis of obstetric indications.

*GLUT1* gene expression under normoxic and hypoxic conditions in the PMSCs is presented in Table 2. The PMSCs from AGA and sIUGR fetuses cultured under normoxic conditions did not exhibit a significant difference in *GLUT1* gene expression (3.2 [1.27] vs. 3.6 [1.97] for AGA-PMSCs and sIUGR-PMSCs, respectively; *p* = 0.059). However, the PMSCs from AGA fetuses cultured under hypoxic conditions exhibited significantly higher *GLUT1* gene expression than did those from sIUGR fetuses cultured under hypoxic conditions (*GLUT1*-FC4: 1.75 [1.22] vs. 1.35 [0.60] for AGA-PMSCs and sIUGR-PMSCs, respectively; *p* = 0.043; *GLUT1*-FC8: 2.15 [1.95] vs. 1.50 [1.45], for AGA-PMSCs and sIUGR-PMSCs, respectively; *p* = 0.043).

*GLUT3* gene expressions in the PMSCs under normoxic and hypoxic conditions for 4 and 8 h are presented in Table 3. Under normoxic conditions, *GLUT3* gene expression was 4.9 [2.1] in the PMSCs from AGA fetuses and 5.8 [1.7] in the PMSCs from sIUGR fetuses; under normoxic conditions, gene expression was higher in the PMSCs from sIUGR fetuses (*p* = 0.042, Wilcoxon signed-rank test). *GLUT3* gene expression under hypoxic conditions was significantly higher in the PMSCs from AGA fetuses than in those from sIUGR fetuses (FC4: 2.0 [1.1] vs. 1.4 [0.75], for AGA-PMSCs and sIUGR-PMSCs, respectively; *p* = 0.043; FC8: 1.24 [1.22] vs. 0.75 [1.05], for AGA-PMSCs and sIUGR-PMSCs, respectively; *p* = 0.027).

HIF-1α protein expression in the PMSCs under normoxic and hypoxic conditions is illustrated in Figure 1. Under normoxic conditions, after culturing under hypoxic conditions for 4 and 8 h, HIF-1α protein expression did not significantly differ between the PMSCs from AGA and sIUGR fetuses (Appendix A).

## 4. Discussion

The results of this study indicate that hypoxia increased the expression of GLUT1 and *GLUT3* in the PMSCs derived from AGA fetuses. This result is consistent with the finding that glucose uptake and transfer by the placenta increased by 13% and 15%, respectively, in animals with maternal inhalation hypoxia [13]. Hypoxia-induced *GLUT1* and *GLUT3* upregulation was higher in the PMSCs from AGA fetuses than in those from sIUGR fetuses. We propose that the placental cells of sIUGR fetuses have a lower hypoxia-induced upregulation of glucose transport than those of AGA fetuses. Furthermore, the decrease in the hypoxia-induced upregulation of GLUTs in the PMSCs from sIUGR fetuses did not result from decreased HIF-1α protein production.

The etiology of IUGR is multifactorial; the etiology of MC twins with sIUGR is mostly attributed to the considerably unequal sharing of the placenta between the twins, which reduces placental blood flow to the sIUGR fetus [9,20]. This reduction in placental blood flow is likely the most common pathway causing IUGR. Because of insufficient blood supply, IUGR fetuses must adapt to an unfavorable environment. Glucose is a primary energy source for the fetus; the fetus must obtain glucose from maternal plasma through the placenta. Because of inadequate passive glucose transport to the fetus, a carrier-mediated transport system is necessary to fulfill fetal needs [2]. Evaluating changes in GLUT expression following the placenta’s response to hypoxia is thus a reasonable strategy for studying IUGR fetal adaptation to hypoxia because studying such changes in vivo is difficult. Hypoxia could increase the *GLUT1* and *GLUT3* expression in placenta mesenchymal stem cells (PMSCs) [6], and hypoxic upregulation of GLUTs in BeWo choriocarcinoma cells was reported to be mediated by HIF-1α [3]. Our HIF-1α protein expression results indicate that the hypoxia-induced upregulation of HIF-1α expression was not decreased in the PMSCs from sIUGR fetuses. The decrease in the hypoxia-induced upregulation of *GLUT1* and *GLUT3* gene expression in the PMSCs derived from sIUGR fetuses likely does not result from decreased HIF-1α expression. Rather, we suspect a decreased response of *GLUT**1* and *GLUT3* upregulation to HIF-1α stimulation in the PMSCs of sIUGR fetuses.

The expression of HIF-1α protein significantly rose at 4 h after hypoxia condition and decreased at 8 h after hypoxia condition in both AGA and sIUGR PMSCs. The HIF-α protein has a short turnover time (half-life of 5 min) and can quickly respond to local O_2_ fluctuations [16]. Thus, the HIF-1α protein could not accumulate from 4 to 8 h after hypoxia conditions in PMSCs. A growing body of evidence supports the crucial effect of oxygen concentration on various types of stem cells. Long-term hypoxia led to decreased gene expression of core transcription factors in human pluripotent stem cells culture in vitro [21]. A combination of the short half-life of the HIF-1α protein and long-term hypoxia may lead to decreased gene expression; 8 h hypoxia may be long enough for the decreasing of the expression of the HIF-1α protein in human PMSCs.

Using an experimental model of an MC twin pregnancy with sIUGR allowed us to investigate the effects of genetics on differences in the hypoxia-induced upregulation of *GLUT1* and *GLUT3* expression in the PMSCs. Intrauterine glucose transfer is regulated by several factors: Glucose supply, placental glucose metabolism, and placental glucose transporter density [2]. Umbilical venous glucose concentrations were reported to be directly related to maternal concentrations [22]. In an MC twin pregnancy with sIUGR, maternal plasma glucose levels for both fetuses were reported to be similar [23]. Therefore, after excluding genetics and differing maternal plasma glucose concentrations, we speculated that placental glucose transporters activity is responsible for the fetal glucose supply in MC twin pregnancies. Therefore, placental glucose transporters gene expression levels served as a means of gauging glucose transport capacity in this study.

The limitations of this study are as follows: First, we used PMSCs to evaluate the glucose transport capacity of the placenta; however, the response of PMSCs to hypoxia may not reflect the placental response to hypoxia in vivo because the placenta has multiple cell components including trophoblasts and mesenchymal cells [24]. In this study, we examined only the effect of hypoxia on the PMSCs. The study of trophoblast cells or placental explants may provide further information on how the human placenta reacts to hypoxia [25]. Second, although *GLUT1* and *GLUT3* are recognized as the chief GLUTs in the placenta, they may not represent the total placental glucose transport capacity. Third, the sample size of this study was small, and the sIUGR fetuses in the included cases exhibited different umbilical artery (UA) Doppler patterns. sIUGR UA Doppler patterns may reflect varying degrees of placental share discordance at delivery [20]. Examination of more cases stratified by an sIUGR UA Doppler pattern may elucidate the effect of hypoxia on the placentas of sIUGR fetuses more clearly.

## 5. Conclusions

We evaluated *GLUT1* and *GLUT3* gene and protein expression levels in PMSCs and determined that the hypoxia-induced upregulation of *GLUT1* and *GLUT3* expression was lower in the PMSCs from sIUGR fetuses than in those from AGA fetuses. This was not because of decreased HIF-1α protein expression in PMSCs from sIUGR fetuses. The placental cells of sIUGR fetuses may have insufficient capacity to increase glucose transport when faced with hypoxia compared with their AGA counterparts.

## Figures and Tables

**Figure 1 genes-13-00752-f001:**
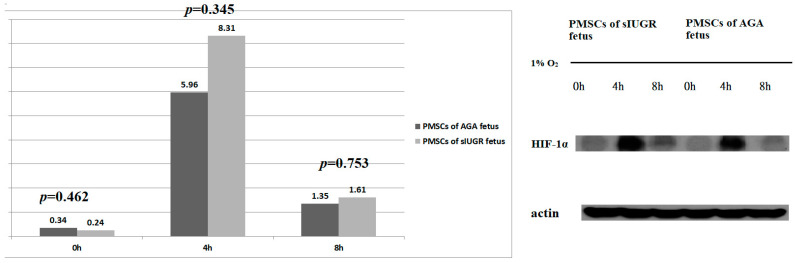
HIF-1α protein expression in placental mesenchymal stem cells (PMSCs) cultured under normoxic conditions and fold changes (FCs) after 4 and 8 h under hypoxic conditions; PMSCs from monochorionic twin fetuses with appropriate-for-gestational-age (AGA) growth and selective intrauterine growth restriction (sIUGR). HIF-1α protein-0: HIF-1α protein expression under normoxic conditions. HIF-1α protein-FC4: HIF-1α protein expression FCs after culturing for 4 h under hypoxic conditions. HIF-1α protein-FC8: HIF-1α protein expression FCs after culturing for 8 h under hypoxic conditions. HIF-1α protein expression under normoxic conditions is expressed as a fraction of actin expression (HIF-1α/actin). HIF-1α protein-FC4 and HIF-1α protein-FC8 are expressed as (protein expression after 4 [8] h of hypoxia) ÷ (protein expression under normoxic conditions). Data are expressed as the median. *p* values were generated using the Wilcoxon signed-rank test.

**Table 1 genes-13-00752-t001:** Characteristics of the six pairs of monochorionic twin pregnancies with selective intrauterine growth restriction (sIUGR).

	Gestational Age of Delivery (Week)	Birth Weight of AGA Fetus (gm)	Birth Weight of sIUGR Fetus (gm)	UA Doppler of sIUGR Fetus at Delivery	Inter-Twin Birth Weight Discordance (%)
Twin 1	27.43	930	395	REDF	57.53
Twin 2	30.14	1360	690	PEDF	49.26
Twin 3	30.43	1350	674	REDF	50.07
Twin 4	34.43	2155	1070	AEDF	50.35
Twin 5	31.00	1535	940	PEDF	38.76
Twin 6	36.57	2680	1730	PEDF	35.45
Median	30.7	1448	815	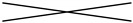	49.70

AGA: appropriate-for-gestational-age. Birth weight discordance was defined as [(birth weight of the AGA twin − birth weight of the sIUGR twin) ÷ (birth weight of the AGA twin)] × 100%. UA: umbilical artery. REDF: reverse of end-diastolic flow. AEDF: absence of end-diastolic flow. PEDF: positive end-diastolic flow.

**Table 2 genes-13-00752-t002:** GLUT1 gene expression fold and fold changes (FCs) in placenta mesenchymal stem cells (PMSCs) derived from the appropriated for gestational age (AGA) fetus’ and selective intrauterine growth restriction (sIUGR) fetus’ placenta territories of monochorionic twins under normal oxygen, after hypoxia culture for four and eight hours.

	*GLUT1*-0	*GLUT1*-FC4	*GLUT1*-FC8
AGA-*GLUT1*	3.2 [1.27]	1.75 [1.22]	2.15 [1.95]
sIUGR-*GLUT1*	3.6 [1.97]	1.35 [0.60]	1.50 [1.45]
*P* *	0.059	0.043	0.043

*GLUT1*-0: *GLUT1* Gene expression at normal oxygen concentration culture. *GLUT1*-FC4: *GLUT1* gene expression fold changes after four hours hypoxia culture. *GLUT1*-FC8: *GLUT1* gene expression fold changes after eight hours hypoxia culture. Gene expression at normal culture are expressed as relative expression comparing to GAPDH expression (*GLUT1*/*GAPDH*). *GLUT1*-FC4 and *GLUT1*-FC8 are expressed as gene expression at four (eight) hours of hypoxia culture/gene expression at normal oxygen culture. Data are expressed as median [inter-quartile range]. *P* *: *p* value of FCs between PMSCs of AGA and sIUGR fetuses at four or eight hours of hypoxia culture. *p* values were generated by Wilcoxon signed-rank test.

**Table 3 genes-13-00752-t003:** *GLUT3* gene expression fold changes (FCs) in placenta mesenchymal stem cells (PMSCs) derived from the appropriated for gestational age (AGA) fetus’ and selective intrauterine growth restriction (sIUGR) fetus’ placenta territories of monochorionic twins under normal oxygen, after hypoxia culture for 4 and 8 h.

	*GLUT3*-0	*GLUT3*-FC4	*GLUT3*-FC8
PMSCs of AGA-*GLUT3*	4.9 [2.1]	2.0 [1.1]	2.2 [2.2]
PMSCs of IUGR-*GLUT3*	5.8 [1.7]	1.4 [0.75]	1.7 [1.9]
*P* *	0.042	0.043	0.027

*GLUT3*-0: *GLUT3* gene expression at normal oxygen concentration culture. *GLUT3*-FC4: *GLUT3* gene expression fold changes after four hours of hypoxia culture. *GLUT3*-FC8: *GLUT3* gene expression fold changes after eight hours of hypoxia culture. Gene expression at normal culture are expressed as relative *GLUT3* gene expression comparing to *GAPDH* gene expression (*GLUT1*/*GAPDH*). *GLUT3*-FC4 and *GLUT3*-FC8 are expressed as gene expression after four (eight) hours of hypoxia/gene expression at normal oxygen culture. Data are expressed as median [inter-quartile range]. *P* *: *p* value of FCs between PMSCs of AGA and sIUGR fetuses after four or eight hours of hypoxia culture. *p* values are generated by Wilcoxon signed-rank test.

## Data Availability

The datasets obtained and/or analyzed during the current study are available from the corresponding author on reasonable request.

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
