# Peer review of "Effect of Hypoxia on Glucose Transporter 1 and 3 Gene Expression in Placental Mesenchymal Stem Cells Derived from Growth-Restricted Fetuses"

_genes, 2022, doi:10.3390/genes13050752_

Round 1
Reviewer 1 Report
I have some concerns regarding MS as listed below:
Abstract:
Page 1 line 15 “How the effect of hypoxia to the gene expression of placenta GLUT1 and GLUT3 in growth restricted fetus is unknown.” -
Page 1 line 20-21 the sentence: “Fold changes in gene expression under hypoxic conditions at 4 (FC4) and 8 (FC8) h were expressed as a fraction of the expression of PMSCs cultured under nonhypoxic conditions” is superfluous. Please, provide here only basic info about techniques used in experiments
There are some abbreviations in Abstract section illegible to the Readers. Full names for the acronyms should be added.
- Introduction
More background and less description of experimental design. The Authors presented experimental design in discussion section. A few more references may be added
- Materials and Methods
Page 1 line 106-107 “After attachment to the well, the cells were cultured under normoxic (20% O2 and 5% CO2) indefinitely or hypoxic conditions (1% O2 and 5% CO2) for 4 or 8 h. “– please provide more information about this experiment, eg equipment, incubators etc.
PMSC characterization – the flow cytometry data should be included to supplementary file. The identification of PMSCs phenotype is a great job and it may be very interesting for the Readers
RNA extraction and real-time qPCR
Is glyceraldehyde-3-phosphate dehydrogenase (GAPDH) the best choice as a reference to study the expression of genes regulating glucose metabolism? Maybe data for actin is available.
- Results
Why western blot for GLUTs was not included into experimental design? It would give more information on the function of the proteins
It would be very informative to show how the concentration of glucose changes during incubation at established time points. If the measurement of glucose uptake is impossible, the measurement of amount of glucose loss from medium would be also suitable. What about acidification of medium? It is an important parameter of the rate of glucose catabolism during hypoxia (and also easy to perform)
- Discussion
The Authors indicate and discuss the possible limitations of the study. Authors should emphasize the novelty of their findings. Some new references should be introduced, possibly referring to the role of transcription factors such as HIFs in regulation of metabolism in cells at the early stages of embryonic development (as reviewed in: doi: 10.3390/jpm11090905). Is it possible based on the obtained results to speculate why the expression of HIF1A rises at 4h and then decreases? Is it important for cellular metabolism?
The conclusions are consistent with the data obtained.
The proper nomenclature for genes and proteins should be introduced throughout the text (capital letters for human genes etc.)
General remarks
The introduction provides some background information, but the Authors focused on introducing the study design that is described again in Discussion section. In my opinion this section gives an opportunity to outline some important metabolic mechanisms that may take place in IUGR placenta and more deeply refer to the great importance of the role of HIFs in the proper/disturbed fetus growth.
The MS is scientifically sound and presented experiments provide a substantial amount of new information. However, the data is a bit scarce – perhaps a classification as short communication rather than original article would be a better choice for this MS?
The proper nomenclature for genes and proteins should be introduced throughout the text (capital letters for human genes, not for proteins etc.)
Please, add captions to the supplementary data
The listed points should be improved. Minor revision is needed
Kind regards,
Author Response
I have some concerns regarding MS as listed below:
Abstract:
Page 1 line 15 “How the effect of hypoxia to the gene expression of placenta GLUT1 and GLUT3 in growth restricted fetus is unknown.” -
Page 1 line 20-21 the sentence: “Fold changes in gene expression under hypoxic conditions at 4 (FC4) and 8 (FC8) h were expressed as a fraction of the expression of PMSCs cultured under nonhypoxic conditions” is superfluous. Please, provide here only basic info about techniques used in experiments
There are some abbreviations in Abstract section illegible to the Readers. Full names for the acronyms should be added.
Answer: Thanks for comments; we have corrected those sentences accordingly.
- Introduction
More background and less description of experimental design. The Authors presented experimental design in discussion section. A few more references may be added
Answer: Thank you very, we try our best to explain the experimental design in the introduction and material method section, we also increase our reference number from 18 to 25.
- Materials and Methods
Page 1 line 106-107 “After attachment to the well, the cells were cultured under normoxic (20% O2 and 5% CO2) indefinitely or hypoxic conditions (1% O2 and 5% CO2) for 4 or 8 h. “– please provide more information about this experiment, eg equipment, incubators etc.
Answer: we add “The hypoxic condition (< 1%) was achieved by flushing the chamber with 5% CO2 and 95% N2. The hypoxia exposure was confirmed with O2 concentration monitoring and elevation of HIF-1α protein abundance with Western blotting.[19]” in the” Isolation and culture of PMSCs” section
PMSC characterization – the flow cytometry data should be included to supplementary file. The identification of PMSCs phenotype is a great job and it may be very interesting for the Readers
Answer:Tthanks for comment, we have added the flow cytometry data in the supplementary files.
RNA extraction and real-time qPCR
Is glyceraldehyde-3-phosphate dehydrogenase (GAPDH) the best choice as a reference to study the expression of genes regulating glucose metabolism? Maybe data for actin is available. BMC Pregnancy Childbirth 2021, 21(1):260.
Answer: previously we found the GLUT1 and GLUT3 gene expression can be detected in monochorionic twin placenta.( BMC Pregnancy Childbirth 2021, 21(1):260.) We used GAPDH as a reference for the gene expression. May be we can try to use “Actin” as reference in next study, due to time limited we cannot offer the GLUT1 And GLUT3 gene expression data using actin as reference before and after hypoxia condition in the reply.
- Results
Why western blot for GLUTs was not included into experimental design? It would give more information on the function of the proteins
Answer: Thanks for comments, we try to do the GLUT1 and GLUT3 WB, but the quality is not good enough, and we have limited PMSCs so we only report the gene expression data in this study. May be we can do the WB later if we have enough PMSCs and with mature skill.
It would be very informative to show how the concentration of glucose changes during incubation at established time points. If the measurement of glucose uptake is impossible, the measurement of amount of glucose loss from medium would be also suitable. What about acidification of medium? It is an important parameter of the rate of glucose catabolism during hypoxia (and also easy to perform)
Answer: This is a great idea, but we may re-do the culture process to obtain the data of concentration of glucose change. As we may try trophoblast in hypoxia in next study, we can check the concentration of glucose changes during incubation in trophoblast study.
- Discussion
The Authors indicate and discuss the possible limitations of the study. Authors should emphasize the novelty of their findings. Some new references should be introduced, possibly referring to the role of transcription factors such as HIFs in regulation of metabolism in cells at the early stages of embryonic development (as reviewed in: doi: 10.3390/jpm11090905). Is it possible based on the obtained results to speculate why the expression of HIF1A rises at 4h and then decreases? Is it important for cellular metabolism?
Answer: we have added “The expression of HIF-1α protein significant rise at 4h after hypoxia condition and decreases at 8h after hypoxia condition both in AGA and sIUGR PMSCs. HIF-α protein has a short turnover time (half-life as 5 min) and can quickly respond to local O2 fluctuations.[15] So HIF-1α protein could not accumulate from 4h to 8h after hypoxia condition in PMSCs. A growing body of evidence supports the crucial effect of oxygen concentration on various types of stem cells. Long term hypoxia lead to decrease gene expression of core transcription factor in human pluripotent stem cells culture in vitro.[19] Combined the short half-life of HIF-1α protein and long term hypoxia may lead to decrease gene expression; 8h hypoxia may be long enough for the decreasing of the expression of HIF-1α protein in human PMSCs.” In the discussion section
The conclusions are consistent with the data obtained.
The proper nomenclature for genes and proteins should be introduced throughout the text (capital letters for human genes etc.)
Answer: thanks you for comments: we check the whole manuscript as make sure that gene symbols are italicized, in upper case and protein symbols are not italicized
General remarks
The introduction provides some background information, but the Authors focused on introducing the study design that is described again in Discussion section. In my opinion this section gives an opportunity to outline some important metabolic mechanisms that may take place in IUGR placenta and more deeply refer to the great importance of the role of HIFs in the proper/disturbed fetus growth.
The MS is scientifically sound and presented experiments provide a substantial amount of new information. However, the data is a bit scarce – perhaps a classification as short communication rather than original article would be a better choice for this MS?
The proper nomenclature for genes and proteins should be introduced throughout the text (capital letters for human genes, not for proteins etc.)
Please, add captions to the supplementary data
Answer: Thanks for comment; we have added the captions to the supplementary data in the revised version.
Reviewer 2 Report
Chang et al, compares hypoxia induced upregulation of GLUT1 and GLUT3 in PMSCs derived from monochorionic twins. The models is very interesting and has much potential. The biggest downside was the very small amount of data and poor quality of the western blots. Interpretation of the data was difficult based on oversaturated Actin bands. Therefore, no true conclusion of how levels of HIF1a protein differed between the AGA and sIUGR groups was possible.
Major
- The oversaturation of the Actin loading control makes quantification of HIF1 changes difficult. Since the western blots are a key portion of the manuscript it makes interpreting the data difficult.
Minor
-AGA abbreviation not defined in the abstract.
-Line 183. The p value is stated as 0.059 were the tables shows 0.59.
-Please remove the decorative background on the supplemental figures.
-The HIF1 bands are difficult to see in the Figure 1. Please increase the contrast of the images.
-Supplemental Material is not called out in the manuscript.
-HIF2 can also influence GLUT expression. Therefore the addition of western data determining HIF2 protein levels may explain the difference in hypoxic GLUT induction between AGA and sIUGR.
Author Response
Chang et al, compares hypoxia induced upregulation of GLUT1 and GLUT3 in PMSCs derived from monochorionic twins. The models is very interesting and has much potential. The biggest downside was the very small amount of data and poor quality of the western blots. Interpretation of the data was difficult based on oversaturated Actin bands. Therefore, no true conclusion of how levels of HIF1a protein differed between the AGA and sIUGR groups was possible.
Major
- The oversaturation of the Actin loading control makes quantification of HIF1 changes difficult. Since the western blots are a key portion of the manuscript it makes interpreting the data difficult.
Answer: We have found some WB pictures of HIF-1α protein with less saturation of Actin band in four of the six cases, and re-upload to the supplementary files.
Minor
-AGA abbreviation not defined in the abstract.
Answer: thanks, we have spell out the AGA as appropriate-for-gestational-age in the whole abstract.
-Line 183. The p value is stated as 0.059 were the tables shows 0.59.
Answer: Thanks for comment, we have corrected the error.
-Please remove the decorative background on the supplemental figures.
Answer: thanks for comment, we have removed the decorative background on the supplemental figures.
-The HIF1 bands are difficult to see in the Figure 1. Please increase the contrast of the images.
Answer: Thanks for comment; we have increased the contrast of the HIF-1α bands.
-Supplemental Material is not called out in the manuscript.
Answer: thanks for comment, we have corrected it accordingly.
-HIF2 can also influence GLUT expression. Therefore the addition of western data determining HIF2 protein levels may explain the difference in hypoxic GLUT induction between AGA and sIUGR.
Answer: Thanks for the important comment. From your comment, we found the HIF-2α is also highly expressed in the human placenta.( Zhao, H.;Wong, R.J.;Stevenson, D.K. The Impact ofHypoxia in EarlyPregnancy onPlacental Cells. Int. J. Mol. Sci. 2021,22, 9675.) Previously we found the up-regulation ofGLUT1 and GLUT3 in these cells under hypoxic conditions was mechanistically demonstrated to be mediated by the hypoxia-inducible transcription factor-1α (HIF-1α) only. (Placenta. 2013 November ; 34(11)) But due to the limitations of revising times and the available of PMSCs, maybe we can measure the PMSCs HIF-2α protein expression in hypoxia condition in next study.
Round 2
Reviewer 2 Report
The authors have adequately addressed all the concerns.